# Aquatic Pollution and Risks to Biodiversity: The Example of Cocaine Effects on the Ovaries of *Anguilla anguilla*

**DOI:** 10.3390/ani12141766

**Published:** 2022-07-10

**Authors:** Mayana Karoline Fontes, Luigi Rosati, Mariana Di Lorenzo, Camilo Dias Seabra Pereira, Luciane Alves Maranho, Vincenza Laforgia, Anna Capaldo

**Affiliations:** 1Institute of Biosciences, Litoral Paulista Campus, São Paulo State University “Júlio de Mesquita Filho,” Praça Infante Dom Henrique s/n, São Vicente 11330-900, Brazil; mayana.fontes@unesp.br; 2Department of Marine Sciences, Federal University of São Paulo, Rua Maria Máximo, 168, Santos 11030-100, Brazil; camilo.seabra@unifesp.br; 3Department of Biology, University of Naples Federico II, Via Cinthia, Edificio 7, 80126 Naples, Italy; luigi.rosati@unina.it (L.R.); mariana.dilorenzo@unina.it (M.D.L.); vincenza.laforgia@unina.it (V.L.); 4Center for Studies on Bioinspired Agro-Environmental Technology (BAT Center), 80055 Portici, Italy; 5Campus Guarujá, Morphofunctional Laboratory, University of Rio Preto-UNAERP, Av. D. Pedro I, 3.300 Enseada Guarujá-SP CEP, Guarujá 11440-003, Brazil; lmaranho@gmail.com

**Keywords:** *Anguilla anguilla*, cocaine, gonadotropins, ovary, oogenesis enzymes

## Abstract

**Simple Summary:**

The increasing consumption of illicit drugs by man has determined the presence of these drugs in waters, both fresh and marine, in almost all the world. Many studies have shown the dangerousness of these substances to aquatic organisms, so the goal of this study was to investigate the effects of cocaine, one of the most used drugs, on the ovaries of the *Anguilla anguilla* eel, a critically endangered edible fish, to understand if this drug could also compromise fish reproduction. Our results show that cocaine may impair reproduction in eels through changes in oocyte morphology, expressions of enzymes crucial for oogenesis, and alteration in the serum levels of cortisol and gonadotropins, which regulate the development and maturation of eel gonads. Though this study was conducted on eels, other species may also suffer similar effects from cocaine exposure; our results, therefore, show that illicit drug use is a serious problem not only for humans but also for the environment.

**Abstract:**

Pollution is one of the main causes of the loss of biodiversity, currently one of the most important environmental problems. Important sources of aquatic pollution are illicit drugs, whose presence in waters is closely related to human consumption; their psychoactive properties and biological activity suggest potential adverse effects on non-target organisms, such as aquatic biota. In this study, we evaluated the effect of an environmentally relevant concentration of cocaine (20 ng L^−1^), an illicit drug widely found in surface waters, on the ovaries of *Anguilla anguilla*, a species critically endangered and able to accumulate cocaine in its tissues following chronic exposure. The following parameters were evaluated: (1) the morphology of the ovaries; (2) the presence and distribution of enzymes involved in oogenesis; (3) serum cortisol, FSH, and LH levels. The eels exposed to cocaine showed a smaller follicular area and a higher percentage of connective tissue than controls (*p* < 0.05), as well as many previtellogenic oocytes compared with controls having numerous fully vitellogenic and early vitellogenic oocytes. In addition, the presence and location of 3β-hydroxysteroid dehydrogenase, 17β-hydroxysteroid dehydrogenase, and P450 aromatase differed in the two groups. Finally, cocaine exposure decreased FSH and LH levels, while it increased cortisol levels. These findings show that even a low environmental concentration of cocaine affects the ovarian morphology and activity of *A. anguilla*, suggesting a potential impact on reproduction in this species.

## 1. Introduction

The progressive loss of biodiversity is currently one of the most important environmental problems, since biodiversity is considered a critical variable for the functioning of the ecosystem services and is probably the most important indicator of the health of the planet [1]. According to most sources, the major direct causes of human-induced biodiversity loss are the fragmentation, degradation, or loss of habitats; the over-exploitation of natural resources; the introduction of non-native species; and climate change and pollution of air and water [2,3,4]. In particular, water pollution is the result of the introduction of various substances into water bodies, which has negative effects on them; in most developed countries, industry is the greatest source of pollution, accounting for more than half the volume of all water pollution and for the deadliest pollutants [5]. However, quite recently, another class of pollutants has been discovered and frequently detected in aquatic environments worldwide, illicit drugs; their presence, together with their metabolites, comes from continuous and increasing drug use by humans. Indeed, these drugs are continuously discharged into aquatic environments and have been detected in municipal sewage treatment plants, surface and drinking water, seawater, marine sediment, and mussels [6,7,8,9,10]. Unlike other pollutants, illicit drugs and their metabolites have psychoactive properties, in addition to having high biological activity; therefore, they can adversely affect not only physiology but also the behavior of aquatic biota [11,12,13]. One of these drugs, cocaine, is the most widely used illicit stimulant drug in Africa, North America, Latin America, the Caribbean, and Europe, with an estimated 19 million users [14]. After consumption, cocaine is rapidly metabolized; a total of 35–54% of the parent compound is hydrolyzed to benzoylecgonine (BE), 32–49% to ecgonine methyl ester (EME), 5% to norcocaine, and only 1–9% of the parent compound is excreted intact [15,16]. Cocaine has been detected in aquatic ecosystems around the world in concentrations ranging from ng L^−1^ to µg L^−1^ [7,9,10,17,18,19]. Once in the environment, cocaine may interact with non-target organisms and cause negative effects. Some in vivo studies [20,21] reported that marine mussels (*Perna perna*) exposed to crack cocaine in concentrations ranging from 5 to 500 µg L^−1^ exhibited DNA damage and cytotoxicity after 48 h and 96 h of exposure, respectively. Crustaceans (*Orconectes rusticus*) injected with cocaine for 3 days (2 and 10 µg L^−1^) showed increased mobility and behavioral changes [22]. In zebrafish larvae, cocaine exposure caused the impairment of skeletal muscle development [23], whereas European eels exposed to environmental cocaine concentrations (20 ng L^−1^) accumulated cocaine in their tissues (including the gonads), causing changes in the endocrine system and skeletal muscles, gills, intestine, skin, liver, and kidneys [24,25,26].

The European eel (*Anguilla anguilla*) is a catadromous fish distributed throughout Europe and North Africa that tolerates a wide range of environmental conditions [27]. After hatching in the Sargasso Sea, *A. anguilla* performs one of the longest migrations in the animal kingdom, traveling approximately 6000 km to the European continent. Several years later, mature eels migrate from fresh water back to the Sargasso Sea to complete their life cycle and die after spawning [28]. European eels have been classified by Union for Conservation of Nature (IUCN) as critically endangered, due to anthropogenic disturbances such as habitat loss and/or degradation, overfishing, environmental changes, and pollution [27,29,30,31]. Indeed, eels build up significant amounts of body fat, thus accumulating and biomagnifying lipophilic and persistent organic pollutants and other chemicals [32,33]. During migration, fat reserves could be catabolized, releasing the stored contaminants into the bloodstream, contaminating, and affecting the reproductive development of the gonads and reproduction itself [29]. 

To date, the effects of cocaine on eel reproduction are not yet known. The aim of this study was to investigate how an environmentally relevant concentration of cocaine (20 ng L^−1^) affected ovarian functions in European eels using morphological, immunohistochemical, and biochemical analyses. 

## 2. Materials and Methods

### 2.1. Animals

Forty-five female specimens of the European eel (*A. anguilla*) at the silver stage of 52.0 ± 3.93 cm and 290.70 ± 36.40 g (mean ± s.d.) were purchased from a fish dealer. Once in the laboratory, the eels were acclimatized for one month in 300 L glass aquaria under natural photoperiod. The water in the aquaria was dechlorinated and well aerated, with the following characteristics: dissolved oxygen, 8.0 ± 0.7 mg L^−1^; ammonia, <0.1 mg L^−1^; salinity, 1°/_oo_; pH, 7.4 ± 0.8; and temperature, 15° ± 1 °C. Every 24 h, all the aquarium water was replaced with fresh tap water; because eels at the silver stage do not eat, they were not fed. This study was carried out in accordance with EU Directive 2010/63/EU for animal experimentation and institutional guidelines for the care and use of laboratory animals and was authorized by General Direction of Animal Health and Veterinary Drugs of the Italian Ministry of Health (Authorization No.221/2015-PR and Authorization No. 22/2015-PR).

### 2.2. Experimental Design

For this fifty-day experiment, a stock solution of 0.006 mg mL^−1^ free-base cocaine (≥97% purity; Sigma-Aldrich Inc., St. Louis, MO, USA) in ethanol was prepared. The eels were randomly divided into three groups (untreated controls, vehicle, and cocaine) of 15 animals each. The experiment was carried out in triplicate; therefore, the specimens in each experimental group were divided into three aquaria containing 5 specimens each. The eels in the cocaine treatment group were exposed to a nominal dose of 20 ng L^−1^ cocaine (1 mL of stock solution, added to the aquaria after water changes every 24 h). Because nearly 90% of cocaine degrades in 24 h at room temperature in water [34], the water removed from the aquaria containing the experimental groups was stored in special containers for three days before being discharged as wastewater. At the same time, three control groups of five eels were only exposed to tap water, and three vehicle groups of five eels were exposed to ethanol at the same concentration as the experimental groups. All groups were kept in 300 L glass aquaria under the conditions described above, and water was changed every 24 h. 

The nominal dose of cocaine (20 ng L^−1^) was selected since this is the average concentration reported for surface water in Italy, China, and Brazil [7,35]. After the exposure period, the eels were anesthetized using MS-222 (ethyl 3-aminobenzoate, methanesulfonic acid salt 98%; Aldrich Chemical Corporation Inc., Milwaukee, WI, USA) at a concentration of 100 mg L^−1^, weighed, measured, and killed by decapitation (Authorization No.221/2015-PR and Authorization No. 22/2015-PR).

Blood was taken from the posterior cardinal vein with a 5 mL syringe to assess hormone levels; after coagulation in Eppendorf tubes for 2–4 h, the blood was centrifuged for 15 min at 2000× *g,* and the serum was collected and stored at −22 °C until the hormone assay was performed. Ovaries were removed from each animal and processed for light microscopy or immunoblot. 

The following parameters were evaluated: (1) general morphology of the ovary, including percentage of connective tissue, and size and number of follicles; (2) presence and distribution of enzymes 3β-hydroxysteroid dehydrogenase (3β-HSD), 17β-hydroxysteroid dehydrogenase (17β-HSD) type 3, and P450 aromatase, which play key roles in oogenesis; (3) serum levels of cortisol, FSH, and LH, involved in the development and maturation of eel gonads.

### 2.3. Histological Analysis

The general morphology of the ovary was evaluated by fixing the ovaries in Bouin’s solution, dehydrating them in graded alcohols, clearing them in Histolemon, and embedding them in Paraplast. Serial 6 µm sections were processed for routine histological analyses and stained with Mallory trichromic stain. A histomorphometric analysis of the ovaries was also performed. Five slides were selected for each ovary; three sections were randomly chosen from each slide, and the area occupied by the connective tissue and the number and area of the ovarian follicles were evaluated. Fifty ovarian follicles from randomly selected areas of different histological sections were analyzed. Histological structures, ovarian follicles, and connective tissue areas were measured with Image J software (National Institutes of Health (NIH)) [36]. All morphological observations were performed using a Zeiss Axioskop microscope (Carl Zeiss MicroImaging s.p.a., Milan, Italy). The images were captured using a camera connected to an IBM computer, with a Kontron Elektronik KS 300 image analysis system (Carl Zeiss) and Adobe Photoshop software.

### 2.4. Immunohistochemical Analysis

#### 2.4.1. Immunoblot

First, antibody specificity was tested via immunoblot analysis. The *Anguilla anguilla* ovaries were homogenized and lysed for 30 min on ice using RIPA lysis buffer containing a mixture of phosphatase and protease inhibitors (Santa Cruz Biotechnology, Milan, Italy). The homogenate was centrifuged at 14,000× *g* for 10 min at 4 °C, and total proteins were measured by Bradford assay (Bio-Rad, Melville, NY, USA) [37]. Briefly, 30 mg of proteins was boiled for 5 min in SDS buffer (50 mM Tris-HCl (pH 6.8), 2 g 100 mL 1% SDS, 10% (*v*/*v*) glycerol, 0.1 g 100 mL 1 Bromophenol blue), separated on 12% SDS-PAGE, and transferred to a PVDF membrane for blotting (Trans-Blot1 Semi-Dry Transfer Cell; Bio-Rad, Milan, Italy) as previously reported [38,39]. The membranes were then incubated for 1 h at room temperature with a blocking buffer (TBS: 0.05% Tween-20 and 5% BSA). Next, the membranes were incubated overnight in the presence of the primary antibody diluted in TBS-T containing 3% BSA at 4 °C: (1) rabbit anti-mouse 3β-HSD (Abcam, Cambridge, UK) diluted to 1:1000; (2) rabbit anti-human 17β-HSD (type 3 isoform; directly involved in the testosterone synthesis) (Abcam) diluted to 1:500; (3) and rabbit anti-P450 aromatase (Elabscience Biotechnology Inc., Houston, TX, USA) diluted to 1:2000. The membranes were washed four times for 10 min in TBS (0.05% Tween-20) before a 1 h incubation with goat anti-rabbit IgG (HRP) (1:2000; Abcam ab-6721) secondary antibody diluted in TBS-T containing 2% BSA. The membranes were washed four other times in TBS, and specific protein bands were detected using diaminobenzidine (DAB; Sigma-Aldrich, Milano, Italy) as a chromogen. Negative controls were performed by omitting the primary antibodies. 

#### 2.4.2. Immunohistochemistry

For immunohistochemical analyses, 5 µm thick sections placed on poly-L-lysine slides (Menzel-Glaser, Braunschweig, Germany) were dewaxed, rehydrated in a graded series of alcohol, and heat-treated (microwave) for 20 min in 10 mM citrate (pH 6.0) antigen-retrieval buffer. The slides were then washed in PBS 1X, treated with 2.5% H_2_O_2_ for 40 min to reduce endogenous peroxidase activity, and blocked for 1 h at room temperature with normal goat serum (Pierce, Rockford, IL, USA) to reduce non-specific background, as described elsewhere [40,41]. Sections were incubated overnight at 4 °C with the following three primary antibodies: rabbit anti-mouse 3β-HSD (1:750, Abcam); rabbit anti-human 17β-HSD (type 3 isoform; directly involved in testosterone synthesis; 1:350, Abcam); and rabbit anti-P450 aromatase (1:150, Elabscience Biotechnology Inc.). The following day, sections were washed in PBS and incubated with HRP-conjugated goat anti-rabbit/mouse secondary antibody diluted to 1:2000 in normal goat serum for 1 h at room temperature. Finally, sections were stained using diaminobenzidine (DAB) as a chromogen and counterstained with Mayer’s hematoxylin. For the negative controls, the primary antibody was omitted from incubation.

### 2.5. Hormone Determination

This study determined the levels of cortisol, follicle-stimulating hormone (FSH), and luteinizing hormone (LH) with an enzyme-linked immunosorbent assay (ELISA, DIAMETRA, Boldon, UK) as previously described [42,43]. The detection limit for cortisol sensitivity was 2.44 ng/mL, with an analytical range of 10–500 ng/mL and incubation time of 60 ± 15 min. The detection limit for FSH sensitivity was 0.17 IU/mL, with an analytical range of 5–100 IU/mL and incubation time of 60 ± 15 min. The detection limit for LH sensitivity was 0.22 IU/mL, with an analytical range of 5.0–200 IU/mL and incubation time of 60 ± 15 min.

### 2.6. Statistical Analysis

The quantitative data were subjected to statistical analysis, and values were expressed as means ± SD. Normality was confirmed for all data, along with homogeneity of variance using Bartlett’s test. The data were compared using one-way analysis of variance (ANOVA), followed by the Tukey–Kramer multiple comparison test. Commercial software (Sigma Stat Version 4.0; SPSS, Systat Software Inc., Palo Alto, CA, USA) was used to perform all statistical analyses, and *p* < 0.05 was considered significant.

## 3. Results

### 3.1. General Morphology

No differences were observed between the vehicle and control specimens, but the ovaries were organized differently in the animals exposed to cocaine. *A. anguilla* is characterized by asynchronous oocyte development; the ovaries contain oocytes in the first growth phase characterized by: absent or single yolk vesicles (previtellogenic oocytes; pvOos); oocytes in which vitellogenesis has begun with yolk vesicle accumulation in one part of the cytoplasm and a nucleus visible at the center (early vitellogenic oocytes; evOos); and large oocytes in which the yolk vesicles have accumulated throughout the cytoplasm and cover the nucleus (fully vitellogenic oocytes; fvOos). All oocytes are present in a follicular structure composed of follicular cells and surrounded by connective cells such as theca cells, which have an endocrine function (Figure 1). The control specimens contained several fvOos and evOos, with little connective tissue between the cells (Figure 1A–D). The ovaries from the animals exposed to cocaine contained more pvOos, with few fvOos and evOos (Figure 1E–H). The morphological observations agreed with the results of the histomorphometric analysis; indeed, the area occupied by the connective tissue was greater in the animals exposed to cocaine (*p* < 0.05) than in control animals (Figure 2A). Moreover, the follicles were smaller in the exposed animals than in the control specimens (*p* < 0.05) (Figure 2B), while the number of follicles was nearly identical in both the control and exposure groups (*p* > 0.05), although the eels exposed to cocaine had slightly more (Figure 2C).

### 3.2. Immunoblot

A Western blot assay performed on protein extracts from *A. anguilla* showed that the antibodies against 3β-HSD, 17β-HSD, and P450 aromatase reacted with the eels’ ovarian proteins. One band of 42 kDa was observed to be positive to rabbit anti-3β-HSD antibody; one band of 35 kDa was positive to rabbit anti-17β-HSD antibody; and one band of 50 kDa was positive to rabbit anti-P450 aromatase antibody. The bands corresponded to the molecular weight of these enzymes, demonstrating the validity of using these antibodies in the ovarian follicles of *A. anguilla* (Appendix A). 

### 3.3. 3β-HSD Localization

The distribution of the immunohistochemistry signals for 3β-HSD, aromatase, and 17β-HSD was similar in the ovarian follicles of the control specimens; a strong and diffuse signal was observed within evOos and fvOos, in the cytoplasm as well as at the edge of the cells. A weak signal was also observed in the follicular and theca cells and in the connective tissue (Figure 3A–C). The expression of 3β-HSD in the ovarian follicles of the exposed specimens was less evident than those of the other enzymes and was also seen on the edge of pvOos and evOos; a weak signal was observed in the connective tissue (Figure 3D–F). No signal was evident in the follicular and theca cells (Figure 3D–F), and no positive reaction was detected in the control section (Figure 3F insert).

### 3.4. 17β-HSD Localization

The immunohistochemistry analysis also revealed a strong signal for 17β-HSD in the control specimens; oocytes were positive to anti-17β-HSD antibody, with a strong signal in fvOos and evOos, whereas a faint signal was registered in the follicular and theca cells as well as in the connective tissue (Figure 4A–C). In contrast, the oocytes from the exposed animals exhibited a less diffuse signal, mainly in pvOos, evOos, follicular cells, and connective tissue. No signal was found in the theca cells (Figure 4D–G), nor was any positive reaction detected in the control section (Figure 4E, insert).

### 3.5. P450 Aromatase Localization

The immunohistochemistry analysis showed the presence of enzyme P450 aromatase in the ovarian follicles from *A. anguilla*. The enzyme showed a strong and widely distributed signal in the control specimens, specifically in the cytoplasm of both fvOos and evOos and in the follicular, theca, and connective cells (Figure 5A–D). Similarly, in the exposed animals the signal of 17β-HSD was visible in the pvOos and evOos as well as in the theca and connective cells, but not in the follicular cells (Figure 5E–G). Figure 5F (insert) demonstrates the lack of signal for P450 aromatase in the negative control section.

### 3.6. Serum LH, FSH, and Cortisol Concentrations

Figure 6 presents the serum levels of LH, FSH, and cortisol. Lower LH and FSH concentrations and higher cortisol concentrations were found in the animals exposed to cocaine compared with the control specimens.

## 4. Discussion

Many studies showed the effects of environmental illicit drugs on fish, for example, the bioaccumulation of illicit drugs and their metabolites in the muscle, liver, and gills of multiple fish species [44]; addiction and behavior alteration induced in brown trout *Salmo trutta* by metamphetamine [45]; dose- and/or time-dependent alterations in reactive oxygen species induction, the activity of antioxidants catalase and superoxide dismutase, and glutathione S-transferase and malondialdehyde contents induced in larvae of medaka fish (*Oryzias latipes*) by ketamine [46]. The adverse effects of cocaine on European eels have already been reported [24,25,47,48,49], but the present study provides the first body evidence indicating that an environmentally relevant concentration of cocaine induces histological and immunohistochemical changes in the gonads of the European eel, supporting our hypothesis that cocaine may impair reproduction in this species through changes in oocyte morphology, expressions of crucial enzymes, and alteration in the serum levels of cortisol and gonadotropins.

Our histological analyses showed that oocytes from the eels exposed to cocaine exhibited a lower maturation rate than those from control specimens. In fact, a greater mean follicle area, a lower percentage of connective tissue, and more frequent presence of evOos (oocytes with small yolk vesicles restricted to the cell cortex) and fvOos (oocytes with enlarged and more abundant yolk vesicles) were observed in the control animals, while more pvOos (few lipid droplets without yolk vesicles) and connective tissue were found in the exposed animals. 

Many studies showed adverse effects on the reproduction of organisms exposed to cocaine. For example, cocaine (10, 20, 40, or 80 mg/kg) affected oocyte development and follicular fluid in rats, decreasing periovulatory serum progesterone [50]; moreover, cocaine (4 mg/kg) disrupted folliculogenesis in female rhesus monkeys due to decreased estradiol levels [51].

In *Drosophila melanogaster*, developmental defects were induced during oogenesis (including aberrant follicle morphogenesis and vitellogenic follicle degeneration) after two weeks of cocaine exposure (0.75 mg/mL; 1.5 mg/mL; 2.0 mg/mL) [52]. To our knowledge, no other studies have investigated the effects of cocaine on fish oogenesis. Our results suggest that cocaine effects could directly affect the eel ovary and/or be mediated via decreased gonadotropin levels; further study is required to clearly identify the mechanisms involved. 

The development and maturation of vertebrate gonads are positively controlled by the activity of several enzymes, such as P450 aromatase, 17β-HSD, and 3β-HSD. We found the presence of these enzymes in European eel ovaries, in agreement with other studies performed in teleost fish. Indeed, 3β-HSD was found in the ovary, testis, and interrenal gonads of rainbow trout (*Oncorhynchus mykiss*) [53]. In gonads of the Japanese eel, P450 aromatase was immunolocalized in the innermost follicular layer [54], while 3β-HSD and 17β-HSD were expressed in the ovary of *Clarias batrachus* [55]. 

All the enzymes investigated in this study play key roles in oogenesis. P450 aromatase catalyzes the conversion of androgens into estrogens, thus regulating sexual differentiation [56], while 17β-HSD controls the last step in the formation of all estrogens and plays a key role in sex steroid biology (including 17β-estradiol, E2) and gametogenesis [57]; moreover, 3β-HSD regulates the production of progesterone [58]. The expressions of enzymes in the gonads have been suggested as key steps in steroidogenesis synthesis, which is crucial for ovarian maturation and female reproduction in teleosts [59,60]. Our finding that European eels exposed to cocaine exhibited a weaker antibody-labeled signal than controls, as well as a different localization of these enzymes, suggests that chronic exposure to cocaine may negatively affect oogenesis and steroidogenesis in these animals. Indeed, P450 aromatase levels rise during maturation, and the expression of this enzyme is a main limiting factor for the synthesis of E2 in the eel ovary, because the 17β-HSD-I gene is highly transcribed at the vitellogenic stage [61]. Furthermore, decreased 3β-HSD affects the conversion of pregnenolone, 17*a*-hydroxypregnenolone, and dehydroepiandrosterone to progesterone, 17*a*-hydroxyprogesterone, and androstenedione, respectively, which are crucial steps in the biosynthesis of sex steroids [62]. Therefore, by considering the physiological role of the enzymes and their involvement in oogenesis, our results suggest that cocaine may affect this process and, in turn, reproductive success in eels, even if more detailed studies are required to confirm this hypothesis.

Furthermore, the development and maturation of eel gonads are positively controlled by two gonadotropins (LH and FSH) and cortisol [63,64,65]. In teleost fish, FSH controls androgen and estrogen synthesis as well as spermatogenesis and oogenesis, while LH is responsible for the production of a progesterone-like hormone, final gamete maturation, and ovulation or sperm release [66].

In *A. australis*, FSH promotes E2 biosynthesis in ovarian follicular cells, while LH regulates final oocyte maturation by stimulating the production of maturation-inducing hormones [67]. Furthermore, cortisol stimulates LH synthesis in the European eel [60]. We found that cocaine exposure decreased both FSH and LH levels, while it increased cortisol levels, as previously observed in eels [48]. Our findings contrast with those of human studies, however, where both FSH and LH levels increased after acute cocaine administration [68]; they are also in contrast with studies of female rhesus monkeys, where cocaine increased LH levels in the presence of low basal E2 levels [69]. This discrepancy could result from differences in type of exposure (chronic vs. acute), animal species, and/or doses administered, but dopamine may also play a role. Cocaine is known to affect the dopamine system, blocking the reuptake of catecholamines (dopamine, norepinephrine) and serotonin, leading to an increased synaptic concentration of these neurotransmitters. Dopamine plays a critical role in the sexual maturation of eels by inhibiting the synthesis and release of gonadotropins [63], resulting in inhibited gonadal development [70]. Previous studies on European eels [48] showed that cocaine increased brain and plasma dopamine levels, so this increase may have caused the decreased synthesis/release of FSH and LH. Although cocaine exposure increased cortisol levels, which should stimulate LH release, dopamine-induced inhibition presumably prevailed over increased cortisol.

## 5. Conclusions

This study provides the body first evidence that even low environmentally relevant concentrations of cocaine induced histological changes in eel ovary morphology, mainly in relation to the maturation of ovarian follicles. Furthermore, cocaine adversely influenced the expression and distribution of important enzymes and hormones involved in oogenesis and steroidogenesis, suggesting a potential risk for the reproductive health of eels. Though this study was conducted on eels, it is not difficult to imagine that these or similar effects may also occur in other species, compromising reproductive fitness and contributing to biodiversity loss. So, an effective policy to combat the production and sale of illicit drugs could be useful not only to man but also to the aquatic biota.

## Figures and Tables

**Figure 1 animals-12-01766-f001:**
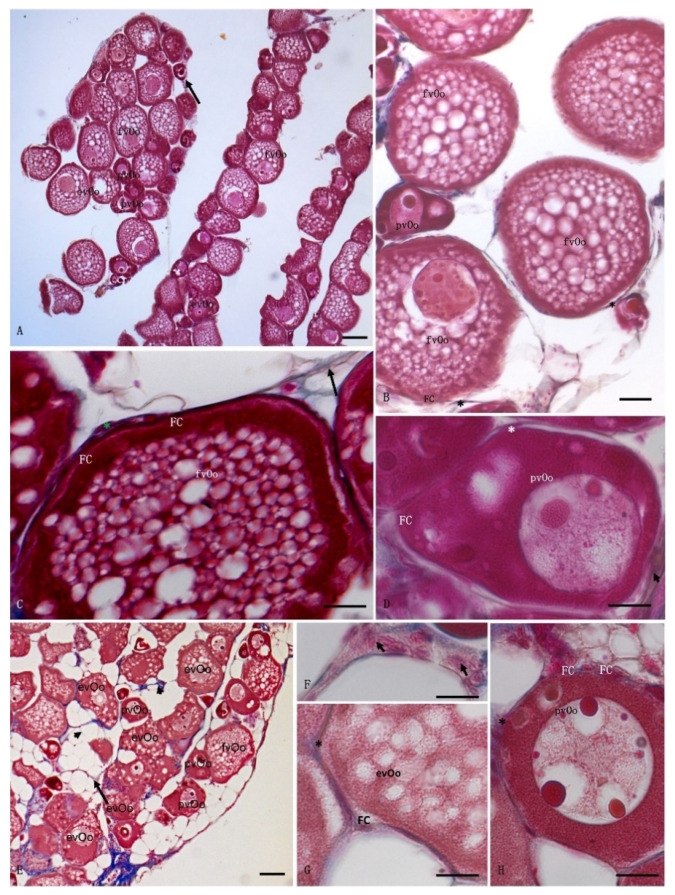
Light micrographs of *Anguilla anguilla* ovaries. Mallory staining. (**A**–**D**) control specimens. The ovary showed numerous fully vitellogenic oocytes (fvOos) and early vitellogenic oocytes (evOos), along with few previtellogenic oocytes (pvOos) and connective interfollicular cells (arrowheads). (**E**–**H**) Exposed specimens, containing more connective tissue (arrows) and previtellogenic oocytes (pvOos) than the control specimens. FC, follicular cells; *, theca cells. Scale bars: (**A**,**E**) = 100 μm; (**B**) = 50 μm; (**C**,**D**,**F**–**H**) = 5 μm.

**Figure 2 animals-12-01766-f002:**
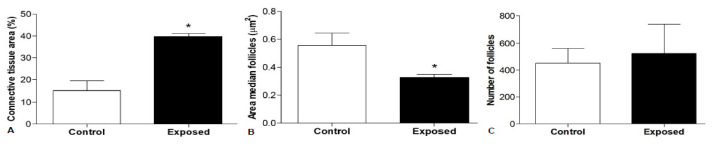
Histomorphometric analysis of *Anguilla anguilla* ovaries. (**A**) Percentage of connective tissue relative to gonad area; in the exposed eels, connective tissue occupied a larger area in the gonads than the control animals. (**B**) Mean area of follicles; follicles in the gonads from the exposed eels were smaller than those in the controls. (**C**) Number of follicles present within the gonads; no differences in the number of follicles were found between the exposed and control specimens. (*) Significantly different (*p* < 0.05) from control values.

**Figure 3 animals-12-01766-f003:**
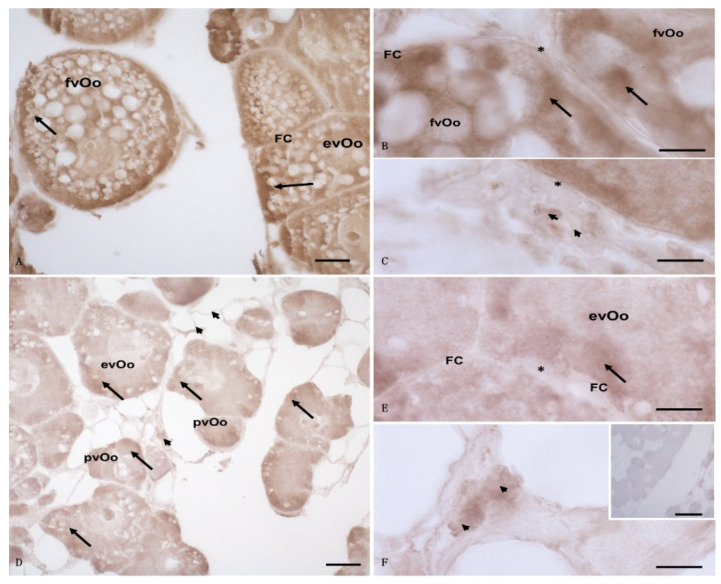
Light micrographs of *Anguilla anguilla* ovaries from control (**A**–**C**) and exposed (**D**–**F**) specimens. Immunolabeling with anti-3β-HSD antibody. (**A**–**C**) a wide and strong positive reaction (arrow) was visible at the level of the fully vitellogenic oocytes (fvOos) and early vitellogenic oocytes (evOos); a weak signal was evident in the follicular (FCs) and theca cells (*) as well as in the connective tissue (arrowheads). (**D**–**F**) Previtellogenic oocytes (pvOos) and early vitellogenic oocytes (evOos) were immunolabeled. In these cells, the immunohistochemical signal was in the cytoplasm as spots (arrow). A weak signal was evident in the connective cells (arrowheads). No positivity was found in the follicular (FCs) or theca (*) cells. (**F**) (insert) Negative control: no labeling was evident in the ovary. Bars: (**A**) = 20 μm; (**B**,**C**,**E**,**F**) = 5 μm, (**D**) = 50 μm, (**F**) (inset) = 100 μm.

**Figure 4 animals-12-01766-f004:**
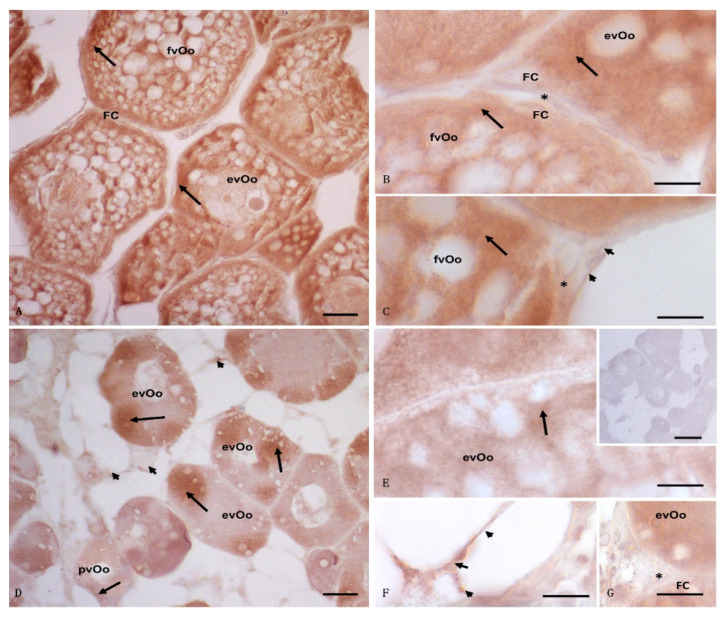
Light micrographs of *Anguilla anguilla* ovaries from control (**A**–**C**) and exposed specimens (**D**–**F**). Immunolabeling with anti-17β-HSD antibody. (**A**–**C**) a wide and strong positive reaction was present in fully vitellogenic oocytes (fvOos) and early vitellogenic oocytes (evOos); a weak signal was evident in the follicular (FCs) and theca (*) cells as well as in the connective cells (arrowheads). (**D**–**G**) early vitellogenic oocytes (evOos) were immunolabeled, and the signal was in some areas of the cytoplasm as spots (arrow); meanwhile, weak positivity (arrow) was localized within previtellogenic oocytes (pvOos), follicular (FC), and connective cells (arrowheads). No signal was evident in the theca cells (*). (**E**) (insert) negative control: no labeling was evident in the ovary. Bars: (**A**) = 20 μm; (**B**,**C**,**E**–**G**) = 5 μm, (**D**) = 50 μm, (**E**) (inset) = 100 μm.

**Figure 5 animals-12-01766-f005:**
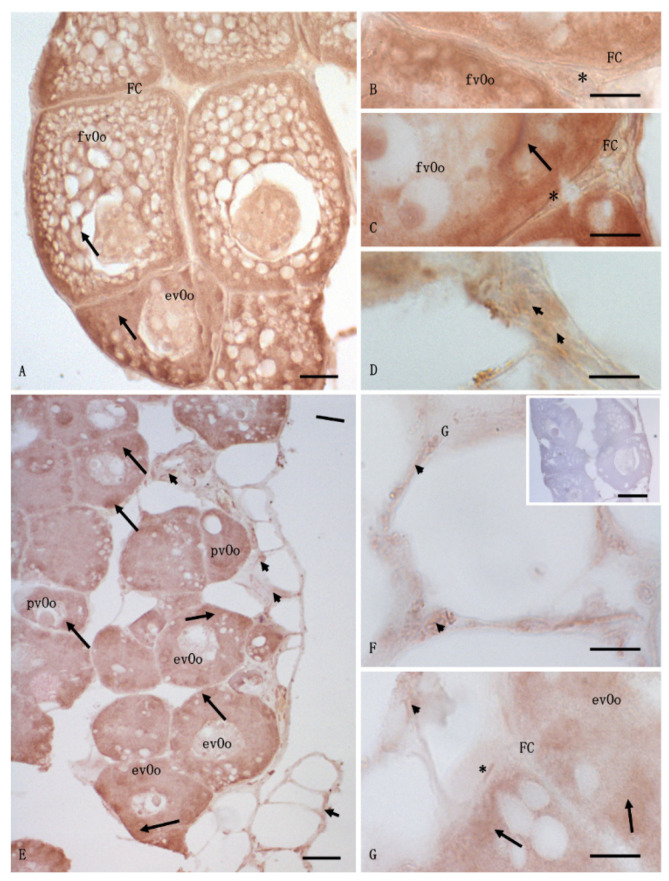
Light micrographs of *Anguilla anguilla* ovaries from control (**A**–**D**) and exposed specimens (**E**–**G**). Immunolabeling with anti-P450 aromatase antibody. (**A**–**D**) a wide and strong positive reaction was present in fully vitellogenic oocytes (fvOos) and early vitellogenic oocytes (evOos), as well as in the follicular (FCs), theca (*), and connective cells (arrowheads). (**E**–**G**) previtellogenic (pvOos) and early vitellogenic oocytes (evOos) were immunolabeled, and the signal was in some areas of the cytoplasm as spots (arrow); positivity was also found in the theca (*) and connective cells (arrowheads), while no signal was seen in the follicular cells (FCs). (**F**) (insert) negative control. No labeling was evident in the ovary. Bars: (**A**) = 20 μm; (**B**–**D**,**F**,**G**) = 5 μm, (**E**) = 50 μm, (**F**) (inset) = 100 μm.

**Figure 6 animals-12-01766-f006:**
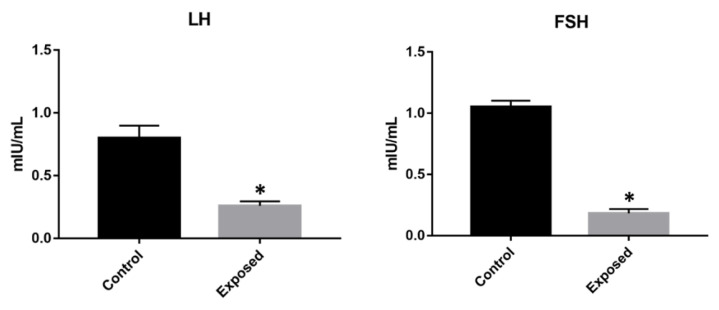
Serum LH, FSH, and cortisol levels in *Anguilla anguilla* control and exposed specimens. Cocaine exposure decreased LH and FSH levels and increased cortisol levels. Values are means ± SD of the mean. (*) serum significantly different (*p* < 0.05) from the control values.

## Data Availability

The data presented in this study are available on request from the corresponding author.

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
