# Peer review of "Aquatic Pollution and Risks to Biodiversity: The Example of Cocaine Effects on the Ovaries of Anguilla anguilla"

_animals, 2022, doi:10.3390/ani12141766_

Round 1

Reviewer 1 Report

The publication presented to me for review concerns one of the most critical areas of human activity - pollution of the aquatic environment with drugs and their metabolites. The work is written in clear and understandable language, the research and descriptive content is presented concretely and to the point, without unnecessary embellishments. I present some minor reservations and comments below. Before the final publication of the research, I expect the final version of the paper to be sent to me.

1. Suggestion to indicate 2-3 publications in the context of the content between lines 47-50, similar to the end of the sentence on line 58.

2. Suggestion for citation in the "Introduction" or "Discussion" section of the publication: "Changes in cadmium concentration in muscles, ovaries and eggs of silver European eel (Anguilla anguilla), during maturation under controlled conditions. Animals, 11 (4), 1027, https://doi.org/10.3390/ani11041027  "

3. The need to remove the first double bracket on line 66.

4. Line 88. Necessity to specify the aim of the work, starting with the words: The aim of the research/study was ...

5.  Lines 91-95. Suggestion to move the content to the "Materials and Methods" chapter.

6.  Lines 95-97. The need to move the content of the last sentence to the chapter "Results" or "Conclusions"

7.  End of line 105. Necessity to make more specific notation of salinity value,

8.  Line 106. Need to explain the term "The water was refreshed every 24 h",% water change. 9.  Chapter "Discussion". Suggestion for more publications on research on fish in the context of cocaine and / or other drugs.

Author Response

Referee 1   Changes in green

The publication presented to me for review concerns one of the most critical areas of human activity - pollution of the aquatic environment with drugs and their metabolites. The work is written in clear and understandable language, the research and descriptive content is presented concretely and to the point, without unnecessary embellishments. I present some minor reservations and comments below. Before the final publication of the research, I expect the final version of the paper to be sent to me.

Suggestion to indicate 2-3 publications in the context of the content between lines 47-50, similar to the end of the sentence on line 58.

DONE

  1. Suggestion for citation in the "Introduction" or "Discussion" section of the publication: "Changes in cadmium concentration in muscles, ovaries and eggs of silver European eel (Anguilla anguilla), during maturation under controlled conditions. Animals, 11 (4), 1027, https://doi.org/10.3390/ani11041027  "

DONE

  1. The need to remove the first double bracket on line 66.

Sorry, I don’t see the double bracket

  1. Line 88. Necessity to specify the aim of the work, starting with the words: The aim of the research/study was

DONE

  1. Lines 91-95. Suggestion to move the content to the "Materials and Methods" chapter.

DONE

  1. Lines 95-97. The need to move the content of the last sentence to the chapter "Results" or "Conclusions"

I deleted the last sentence

  1. End of line 105. Necessity to make more specific notation of salinity value,

DONE

  1. Line 106. Need to explain the term "The water was refreshed every 24 h", % water change.

DONE

  1. Chapter "Discussion". Suggestion for more publications on research on fish in the context of cocaine and / or other drugs.

DONE

Reviewer 2 Report

Major comments

1. line 284: In the legend of Figure 4: Explanation of Figure 4 G is missing

2. line 302: In the legend of Figure 5: Explanation of Figure 5 G is missing

Minor editorial revisions will be needed.

1. line 60: delete and before therefore    add comma after therefore

2. line 65: observed should be detected

3. lines 67-68: in vivo should be italic?   follow the journal style

4. line 106: delete C after 15

5. line 116: add a comma after therefore

6. line 148: delete MicroImaging s.p.a., Milan Italy

7. line 151: Anguilla anguilla should be A. anguilla

8. line 155: Add USA after NY,

9. line 158: add city, state, and country after Bio-Rad

10. line 162: add city, state, country after Abcam

11. line 165: add city, state, country after Biotechnology Inc.

12. line 190: add city, state, country after DAMETRA

13. line 193: minutes should be min

14. line 201: add city, state, country after SPSS

15. line 308: Figure 6: The order of figures should be LH, FSH and Cortisol.  and the Figure legend needs to be revised accordingly

16. line 333: Drosophila melanogaster should be italic

17. line 347: estrogenic hormones should be estrogens

18. line 349: delete a space before -estradiol

19. lines 359, 360: 17a-hydroxy.... should be 17a-hydroxy....

20. line 371: 17b-estradiol (E2) should be E2

21. line 378: estradiol should be E2

Author Response

Referee 2 Changes in blue

Major comments

  1. line 284: In the legend of Figure 4: Explanation of Figure 4 G is missing,

DONE

  1. line 302: In the legend of Figure 5: Explanation of Figure 5 G is missing

DONE

Minor editorial revisions will be needed.

  1. line 60: delete and before therefore    add comma after therefore

Done   

  1. line 65: observed should be detected

Done

  1. lines 67-68: in vivo should be italic?   follow the journal style

Done

  1. line 106: delete C after 15

Done

  1. line 116: add a comma after therefore

Done

  1. line 148: delete MicroImaging s.p.a., Milan Italy

Done (line 149)

  1. line 151: Anguilla anguilla should be AAnguilla

Done (line 152)

  1. line 155: Add USA after NY,

Done (line 156)

  1. line 158: add city, state, and country after Bio-Rad

Done

  1. line 162: add city, state, country after Abcam

Done

  1. line 165: add city, state, country after Biotechnology Inc.

Done

  1. line 190: add city, state, country after DAMETRA

Done

  1. line 193: minutes should be min

Done (line 194)

  1. line 201: add city, state, country after SPSS

The software is provided by University Federico II

  1. line 308: Figure 6: The order of figures should be LH, FSH and Cortisol.  and the Figure legend needs to be revised accordingly

Done (line 310)

  1. line 333: Drosophila melanogaster should be italic

Done

  1. line 347: estrogenic hormones should be estrogens

Done

  1. line 349: delete a space before -estradiol

Done

  1. lines 359, 360: 17a-hydroxy.... should be 17a-hydroxy...

Done (I put the letter a in italics, so I understood)

  1. line 371: 17b-estradiol (E2) should be E2

Done (line 370)

  1. line 378: estradiol should be E2

Done (line 377)

Reviewer 3 Report

Comments on the manuscript:

“Aquatic pollution and risks to biodiversity: the example of cocaine effects on the ovaries of Anguilla anguilla

As a result of their increased use, many illicit drugs are present in fresh and marine waters. This presence is dangerous for aquatic organisms. The effects of these drugs are not perfectly known. The work presented here concerns the effects of cocaine, one of these drugs among the most widely used, on the female genitalia of eels, a fish with long migrations, recognized as an endangered species by the Union for Conservation of Nature (IUCN).

This work is well done and brings useful elements. It could be published after some improvements to the manuscript. Here are some remarks.

Page 2, line 66. “ ([4, 6, 7; 14-16].”: delete “(”.

Page 3, lines 129-131. “the eels were anesthetized using MS-222 (ethyl 3-aminobenzoate, methanesulfonic acid salt 98%, Aldrich Chemical Corporation Inc., Milwaukee, WI, USA) at a concentration of 100 mg L-1, weighed, measured, and killed by decapitation.”: give also here the reference of authorization.

Page 3, line 143-144. “the area occupied by the connective tissue and number and area of the ovarian follicles were evaluated”: give a brief description of the method used to evaluate the different areas.

Page 4, line 151: use italics to write Anguilla anguilla.

Page 4, line 190. “as described previously”: give a brief description of the method used.

Page 6, figure 1. the difference between arrows and arrowheads is not very clear: lengthen the arrows (like in figure 3).

Page 6, figure 1. Previtellogenic oocytes are indicated by “pOo” in the photos instead of “PvOo” in the legend and in the text. Please correct and check if the abbreviations are the same throughout the text.

Page 7, line 244-245. “and one band of 50 kDa positive to rabbit anti-P450 aromatase antibody (Fig. 3).”: Figure 3 is not an immunoblot but an immunohistochemical stain. Figure S1 and Figure 3 given apart from the main text are the same. Please correct.

Page 7, line 248: write “3.3. 3β-HSD localization” instead of “3.3.3β-. HSD localization”.

Page 8, figure 3, page 9, figure 4 and page 10, figure 5: like for figure 1, previtellogenic oocytes are indicated by “pOo” in the photos instead of “PvOo” in the legend and in the text. Please correct and check if the abbreviations are the same throughout the text.

Page 8, line 268: write “ 3.4. 17β-HSD localization” instead of “ 3.4.17β-. HSD localization”.

Page 11, lines 315-316. “([21, 22; 41-43]”: delete “(”.

Page 11, line 332. “([45].”: delete “(”.

References:

Page 15, line 528 : use italics to write “Anguilla”.

Author Response

Referee 3 Changes in red

Comments on the manuscript:

“Aquatic pollution and risks to biodiversity: the example of cocaine effects on the ovaries of Anguilla anguilla

As a result of their increased use, many illicit drugs are present in fresh and marine waters. This presence is dangerous for aquatic organisms. The effects of these drugs are not perfectly known. The work presented here concerns the effects of cocaine, one of these drugs among the most widely used, on the female genitalia of eels, a fish with long migrations, recognized as an endangered species by the Union for Conservation of Nature (IUCN).

This work is well done and brings useful elements. It could be published after some improvements to the manuscript. Here are some remarks.

Page 2, line 66. “ ([4, 6, 7; 14-16].”: delete “(”.

Done

Page 3, lines 129-131. “the eels were anesthetized using MS-222 (ethyl 3-aminobenzoate, methanesulfonic acid salt 98%, Aldrich Chemical Corporation Inc., Milwaukee, WI, USA) at a concentration of 100 mg L-1, weighed, measured, and killed by decapitation.”: give also here the reference of authorization.

Done

Page 3, line 143-144. “the area occupied by the connective tissue and number and area of the ovarian follicles were evaluated”: give a brief description of the method used to evaluate the different areas.

DONE

Page 4, line 151: use italics to write Anguilla anguilla.

Done (suggestion of reviewer 2)

Page 4, line 190. “as described previously”: give a brief description of the method used.

We followed the manufacturer’s instructions

Page 6, figure 1. the difference between arrows and arrowheads is not very clear: lengthen the arrows (like in figure 3).

Done

Page 6, figure 1. Previtellogenic oocytes are indicated by “pOo” in the photos instead of “PvOo” in the legend and in the text. Please correct and check if the abbreviations are the same throughout the text.

Done

Page 7, line 244-245. “and one band of 50 kDa positive to rabbit anti-P450 aromatase antibody (Fig. 3).”: Figure 3 is not an immunoblot but an immunohistochemical stain. Figure S1 and Figure 3 given apart from the main text are the same. Please correct.

Done

Page 7, line 248: write “3.3. 3β-HSD localization” instead of “3.3.3β-. HSD localization”.

Done

Page 8, figure 3, page 9, figure 4 and page 10, figure 5: like for figure 1, previtellogenic oocytes are indicated by “pOo” in the photos instead of “PvOo” in the legend and in the text. Please correct and check if the abbreviations are the same throughout the text.

Done

Page 8, line 268: write “ 3.4. 17β-HSD localization” instead of “ 3.4.17β-. HSD localization”.

Done

Page 11, lines 315-316. “([21, 22; 41-43]”: delete “(”.

Done

Page 11, line 332. “([45].”: delete “(”.

Done

References:

Page 15, line 528: use italics to write “Anguilla”.

Done
